# Relationship between Ground Reaction Forces and Morpho- Metric Measures in Two Different Canine Phenotypes Using Regression Analysis

**DOI:** 10.3390/vetsci9070325

**Published:** 2022-06-28

**Authors:** Giovanni Della Valle, Chiara Caterino, Federica Aragosa, Caterina Balestriere, Alfonso Piscitelli, Cristina Di Palma, Maria Pia Pasolini, Gerardo Fatone

**Affiliations:** 1Department of Veterinary Medicine and Animal Production, University of Naples “Federico II”, 80137 Naples, Italy; giovanni.dellavalle@unina.it (G.D.V.); chiara.caterino@unina.it (C.C.); cristina.dipalma@unina.it (C.D.P.); pasolini@unina.it (M.P.P.); fatone@unina.it (G.F.); 2Centro Traumatologico Ortopedico Veterinario, 16011 Arenzano, Italy; caterinabalestriere@gmail.com; 3Department of Agricultural Sciences, University of Naples “Federico II”, 80056 Portici, Italy; alfonso.piscitelli@unina.it

**Keywords:** GFRS, dolichomorph, mesomorph, peak vertical force, vertical impulse, stance time, withers height, trunk length, dog

## Abstract

**Simple summary:**

The force plate gait analysis is considered the gold standard for the objective assessment of limb function in dogs. Force plate analysis has been employed in several studies using a cohort of healthy dogs as a control group. However, these studies do not consider the subject variability within the same breed. Moreover, the lack of a rigorous analysis of morphometric variabilities in the same breed makes these evaluations poorly reliable. This prospective study aims to investigate the relationship between the ground reaction forces obtained by the force plate gait analysis and the morphometric measures in two different domestic dogs’ morphotypes. Our results highlighted how the ground reaction forces are influenced by morphometric measures not so much as a single contribution, but through the interaction between them. Indeed, the interaction between body weight, withers height, and velocity significantly influenced ground reaction forces with a greater unit increase for mesomorphs. Statistical models used in the available literature only partially explain the influence of morphometric measures on ground reaction forces, and the comparison between dogs should be made not referring to the breeds but the canine morphotype.

**Abstract:**

Force plate analysis assesses gait symmetry and limb loading. However, as previously described, individual and breed variability (body size and conformation) is related to breeding, body conformation, and size. This prospective study aimed to evaluate the influence of morphometric measures on the speed (V), peak of vertical force (PVF), vertical impulse (VI), and stance time (ST) in healthy dolichomorph and mesomorph dogs and their combined effect on and interactions with V, PVF, VI, and ST in the same morphological types. Fifty dogs were enrolled in the current study, and specific morphometric measurements were recorded for each dog. A force platform was used to record the ground reaction forces (GFRs), including PVF and VI. Multiple linear regression models were used for the study purposes. According to our results, GFRs are influenced by morphometric measures (body weight, withers height, and speed) not so much as a single contribution, but by the interaction between them. It is not possible to compare GFRs in dogs that do not belong to the same breed. However, the subjective variabilities make this comparison difficult and poorly reliable. According to the author, the comparison should be made between canine morphological types rather than breeds.

## 1. Introduction

Force plate gait analysis is considered the gold standard for the objective assessment of limb function in dogs. The ground reaction forces (GRF) have been commonly collected to evaluate the outcome of medical and surgical treatment in patients with musculoskeletal disease [1]. 

Domestic animals display a large variability not only in size but also in body conformation [2]. The effects of morphometric variables (body weight and humerus, femur, and paw length) on GRFs were first described by Budsberg in 1987 in a group of 17 healthy dogs [3]. As highlighted in this study, the morphometric measures have a direct correlation to vertical impulse (VI) and a negative linear correlation to peak vertical force (PVF) [3]. In 2011, in a study on 54 dogs of 7 different breeds, Voss and colleagues demonstrated that GRFs are influenced by conformation and body size (BS). Moreover, the author concluded that the group comparisons should only be made when the groups consist of breeds with similar body conformations, making a comparison with a control group not always reliable [1]. To minimise variabilities, current guidelines recommend normalising GRF to body weight (*BW*) and using a narrow velocity (V) range [4,5]. However, significant variability persists in the data of gait analysis despite the normalisation of GRFs [1,2,6]. Several studies investigated and proposed rescaling the gait parameters to *BW* or size, or both combined, as well as the use of relative velocity, based on the theory of dynamic similarity [1,2,6,7]. Even still, after full normalisation to *BW* and BS, force plate data for dogs of different breeds are not necessarily comparable [1,2,8].

Nevertheless, force plate analysis has been applied in several studies using a cohort of healthy dogs as a control group [5,6,7,9]. However, the lack of a rigorous analysis of morphometric variabilities in the same breed makes these evaluations poorly reliable. Voss and colleagues (2010) demonstrated the inherent relationship between *BW*, BS, velocity, and GRFs in 129 dogs; despite full normalisation of GRFs, a small persistent dependency on BS was detected [2]. Similarly, Mölsä and colleagues in 2010 [6] pointed out that the significant differences between the normalised GFRs of two groups of healthy dogs (Labrador and Rottweiler) were attributable to differences in conformation and *BW* between the breeds. Starting from this literature background, we hypothesised that the morphometric variables have a combined effect on GRFs and do not act as a single variable.

This perspective study aimed to **(a)** assess the impact of morphometric measures on V, PVF, VI, and ST in healthy dolichomorphic and mesomorphic dogs; **(b)** evaluate the combined effect of morphometric measurements and their possible interactions on V, PVF, VI, ST; and **(c)** identify differences in these relationships in the two morphological groups considered.

## 2. Materials and Methods

The population data for this perspective study consisted of healthy dogs of different breeds recognised by the Fédération Cynologique Internationale (FCI) and categorized into dolichomorph and mesomorph [10,11]. The morphotypes were defined < t, considering the previously described relationship between height at withers and thoracic conformation [10,11]. The sample recruitment took place at several FCI-approved dog shows as well as at the Veterinary Teaching Hospital (VTH) of the Department of Veterinary Medicine and Animal Production at the University of Naples “Federico II”.

Inclusion criteria were as follows: individuals belonging to the breeds included in the morphological classification, being over 1-year-old, weighing more than 14 kg, and absence of any previous or detectable orthopaedic pathologies at the time of enrollment.

Exclusion criteria were as follows: excessively restless dogs or aggressive temperament, poor leash behaviour, and failure to complete the required number of valid trials within the time frame.

Dogs were assigned to 1 of 2 groups based on two morphological types: dolichomorphs and mesomorphs [10,11].

Breed, sex, age (years), weight (kg), and specific morphometric measurements were recorded for each dog.

### 2.1. Morphometric Measurements

The withers height (*WH*) and trunk length (*L*) of each enrolled dog were measured with a tape measure designed for use in dog shows and competitions; each dog had to remain in standing position. 

The withers are the highest point at the back, starting from the neck base and ending between the scapulae; the length of the trunk was considered as the distance between the spinous process of the second thoracic vertebra and that of the seventh lumbar vertebra.

Similarly, the length of the humerus, represented as the distance between the greater tubercle and the lateral epicondyle of the humerus (O), and the length of the femur (F), as the distance between the greater trochanter and the lateral epicondyle, were measured. All measurements were expressed in centimetres (cm).

### 2.2. Force Gait Plate Analysis

To record GRFs, a 40 × 40 cm platform (PASPORT Force Platform, PS-2141, PASCO Scientific, Roseville, CA, USA) was perfectly allocated in a 4 m walkway platform to avoid the “step effect”.

Before data collection, dogs were allowed to walk freely across the walkway for at least 15 min to familiarise themselves with the environment and the operators. 

A variable number of trials were performed for each subject to obtain a minimum of 3 valid measurements for each limb. 

Each trial was considered valid when the test limb was fully struck on the platform without touching the edges, with the dog walking alongside the handler, without distraction from the surroundings, at a speed between 1 and 1.3 m/s. 

The dog’s velocity was recorded with an advance detector (Motion Sensor II, CI-6742, PASCO Scientific, Roseville, CA, USA), and only trials with a velocity of 1–1.3 m/s were accepted [12]. 

In addition, the forelimb survey was conducted and followed by the ipsilateral hindlimb survey during the same trial. All surveys in which the forelimb and hindlimb curves overlapped or joined were discarded. The maximum time allowed to obtain the three valid trials for each limb was 30 min, and the total number of trials performed was recorded for each subject. Dogs were walked in both directions with a standardised starting position. 

Kinetic GFRs were collected for each limb and included the peak of vertical force (PVF) and vertical impulse (VI), stance time (ST), and speed (V, in m/s). PVF (in N) was defined as the maximum force exerted perpendicular to the surface during the stance phase, while VI (in Ns) was calculated as the area under the vertical force curve. The stance time (in seconds) represented the start of the stance phase until the moment the limb was lifted entirely from the platform. 

The speed was normalised using the formula V* = V/(g*WH*)1/2; where V*, called Froude’s number, defines the relative speed, g represents the gravitational acceleration expressed by a constant of 9.8 m/s^2,^, and *WH* represents the height at the withers of the subject, which in this formula is expressed in meters [2]. 

### 2.3. Statistical Analysis

All data were recorded using spreadsheet software (Microsoft^®^ Excel^®^ 2011) and imported into a statistical analysis software package (IBM^®^ SPSS^®^ Statistics Version 26.0, IBM Corporation, Armonk, NY, USA). We preferred not to use the normalised GRF measures (PVF%, *BW*, VI*, and ST*) in the analyses because our goal was not to compare PVF, VI, and ST values across subjects but to assess the dependency structure between the GRF measures and the morphometric measures.

Therefore, we estimated six different models based on six stepwise regression selections out of those predictors: a canine morphotype dummy variable (*D*); age (*A*); body weight (*BW*); withers height (*WH*); trunk length (*L*); femur (*F*); average speed (*V*); interaction between *D·BW* (Int_D_BW); interaction between *D·WH* (Int_D_WH); interaction between *D·L* (Int_D_L); interaction between *D*·*F* (Int_D_F); interaction between *D*·*V* (Int_D_V); and interaction between *BW·WH* (Int_BW_WH). Humeral length was not included in the regression model as a predictor variable due to the highest collinearity with *WH*. *D* is a binary indicator function, a so-called dummy variable, to alter intercept and/or slope coefficients in the regression model (*D* = 1 if the observation lies in the mesomorphic group; *D*=0 if the observation lies in the dolichomorphic group). 

Thus, a dummy variable was used to model canine morphotype differences on GRF measures, while interactions of *D* with other predictors highlight the differences between dolichomorphic and mesomorphic groups. The regression models with all predictors considered, for both thoracic and pelvic limbs, are shown below:*Y_T* = *B_0* + *B_1∙D* + *B_2∙A* + *B_3∙BW* + *B_4∙WH* + *B_5∙L* + *B_6∙V* + *B_7∙D∙BW* + *B_8∙D∙WH* + *B_9∙D∙L* + *B_10∙D∙V* + *B_11∙BW∙WH* + *ε*(1)
*Y_P* = *B_0* + *B_1∙D* + *B_2∙A* + *B_3∙BW* + *B_4∙WH* + *B_5∙L* + *B_6∙F* + *B_7∙V* + *B_8∙D∙BW* + *B_9∙D∙WH* + *B_10∙D∙L* + *B_11∙D∙F* + *B_12∙D∙V* + *B_13∙BW∙WH* + *ε*
(2)

To the models specified in Equations (1) and (2), we performed backwards stepwise regression that involves starting with all predictors and testing them one by one for statistical significance, deleting any that are not significant. In our analysis, step by step, the algorithm deleted from the model the predictors with *p*-value > 0.1 and therefore statistically not significant. The final step of the backwards regression identifies the best explanatory independent variables. 

## 3. Results

Fifty dogs met the inclusion criteria: 14 were dolichomorphs, and 36 were mesomorphs. Out of the 14 dolichomorphs, 5 were females and 9 males; from 36 mesomorphs, 20 were females and 16 males (Table 1). 

Therefore, the total population examined included 25 females and 25 males. 

The mean ± standard deviation (SD) for age and weight in the dolichomorphs group was 4.6 ± 2.79 years and 37.46 ± 22.04 kg, respectively (Table 2). 

Meanwhile, in the mesomorph group, the mean ± standard deviation (SD) for age and weight were 2.48 ± 1.46 years and 33.35 ± 38.01 kg, respectively (Table 3).

The morphometric measurements expressed as median (range) are shown in Table 4.

The average number of trials required to obtain data for inclusion in the study was 22.08 ± 4.36 for mesomorphs and 17.71 ± 3.65 for dolichomorphs, respectively.

### 3.1. Multiple Regression Analysis

#### 3.1.1. Thoracic PVF

The coefficients of the model selected by stepwise regression for thoracic PVF (*T_PVF*) as a dependent variable are presented in Table 5. The high value of statistic F = 184.9 and the corresponding level of the *p*-value (*p* < 0.001) confirm a statistically significant linear relationship. The adjusted R^2^ coefficient is high (R^2^ = 0.949); it represents the contribution of the set of predictors to explaining the variability in the *T_PVF*.

According to the coefficients in Table 5, the regression equation takes the following form (Model 1):T_PVF=55.32−344.25·D+2.19·BW+1.14·D·BW+289.62·D·V+0.03·BW·WH

The body weight plays a significant role (*p* = 0.032) in explaining *T_PVF*, with an average increase of 2.19 N for each kg. Canine morphology appears discriminating (*p* = 0.015), with mesomorphs showing an average thoracic PVF value lower than dolichomorphs. Canine morphology also has a moderating effect when combined with both body weight (*p* = 0.011) and speed (*p* = 0.016), with mesomorphic subjects showing an average thoracic PVF value of 289.62 N higher than dolichomorphic for each m/s.

Finally, in Model 1, the interaction between body weight and withers height also shows a statistically significant effect (*p* = 0.003).

To compare actual data with the model’s predictions, we can visualise the correlation scatterplot of *T_PVF* for the observed values predicted by the model. As shown in Figure 1, the obtained model proves a very good fit for the actual values.

#### 3.1.2. Thoracic VI

The coefficients of the model selected by stepwise regression for thoracic VI (T_VI) as a dependent variable are presented in Table 6. The high value of statistic F = 291.11 and the corresponding level of the *p*-value (*p* < 0.000) confirm a statistically significant linear relationship. 

The adjusted R^2^ coefficient is high (R^2^ = 0.959); it represents the contribution of the set of predictors to explaining the variability in the T_VI. 

According to the coefficients in Table 6, the regression equation takes the following form (Model 2):$$T_{VI}=6.77+0.65\cdot D\cdot BW-0.72\cdot D\cdot WH+0.54\cdot D\cdot L+0.03\cdot BW\cdot WH$$

The interaction between *BWWH* shows a statistically significant effect on VI (*p* = 0.000). The mesomorph type has a moderating effect when combined with *BW* (*p* = 0.003) and *WH* (*p* = 0.001), showing an average increase of 0.65 N/s for each Kg and an average decrease of 0.72 N/s for each cm of *WH*. 

To compare actual data with the model’s predictions, we can visualise the correlation scatterplot of T_VI for the observed values predicted by the model. As shown in Figure 2, the obtained model proves a very good fit for the real values.

#### 3.1.3. Thoracic ST

The coefficients of the model selected by stepwise regression for thoracic ST (T_ST) as a dependent variable are presented in Table 7. The value of statistic F = 20.96 and the corresponding level of the *p*-value (*p* < 0.001) confirm a statistically significant linear relationship. 

The obtained adjusted coefficient of determination, R^2^ = 0.550, proves a moderate fit of the model to the real data of T_ST.

According to the coefficients in Table 7, the regression equation takes the following form (Model 3):$$T_{ST}=0.083+0.007\cdot WH+0.003\cdot D\cdot BW-0.002\cdot D\cdot WH$$

The *WH* for both groups significantly affects the ST (*p* = 0.000) even with an increased minimal average. Moreover, only for the mesomorphs, *BW* has an average increase of 0.003 s per Kg; at the same time, *WH* has an average decrease of 0.002 s for each cm of *WH*. To compare actual data with the model’s predictions, we can visualise the correlation scatterplot of T_ST for the observed values predicted by the model. As shown in Figure 3, the obtained model proves a moderate fit for the real values.

#### 3.1.4. Pelvic PVF

The coefficients of the model selected by stepwise regression for pelvic PVF (*P_PVF*) as a dependent variable are presented in Table 8. The high value of statistic F = 126.7 and the corresponding level of the *p*-value (*p* < 0.001) confirm a statistically significant linear relationship. The adjusted R^2^ coefficient is high (R^2^ = 0.911); it represents the contribution of the set of predictors to explaining the variability in the *P_PVF*.

According to the coefficients in Table 8, the regression equation takes the following form (Model 4):P_PVF=27.658−385.56·D+2.52·BW+336.132·D·V+0.014·BW·WH

The body weight plays a significant role (*p* = 0.000) in explaining *P_PVF*, with an average increase of 2.51 N for each kg. Canine morphotype appears to be discriminating (*p* = 0.002), with the mesomorph type showing an average pelvic PVF value lower than the dolichomorphic type. Canine morphotype also has a moderating effect when combined with speed (*p* = 0.002), with mesomorphic subjects showing an average pelvic PVF value of 336,132 N higher than dolichomorphic for each m/s. 

To compare actual data with the model’s predictions, we can visualise the correlation scatterplot of *P_PVF* for the observed values predicted by the model. As shown in Figure 4, the obtained model proves a very good fit for the real values.

#### 3.1.5. Pelvic VI

The coefficients of the model selected by stepwise regression for pelvic VI (P_VI) as a dependent variable are presented in Table 9. The high value of statistic F = 648.53 and the corresponding level of the *p*-value (*p* < 0.001) confirm a statistically significant linear relationship. The adjusted R^2^ coefficient is high (R^2^ = 0.964); it represents the contribution of the set of predictors to explaining the variability in the P_VI.

According to the coefficients in Table 9, the regression equation takes the following form (Model 5):$$P_{VI}=-13.488+2.198\cdot BW-0.183\cdot D\cdot WH$$

The body weight plays a significant role (*p* = 0.000) in explaining P_VI, with an average increase of 2.12 N/s for each kg. The *WH* shows an average decrease of 0.183 N/s for each cm of *WH* in the mesomorphs. 

To compare actual data with the model’s predictions, we can visualise the correlation scatterplot of P_VI for the observed values predicted by the model. As shown in Figure 5, the obtained model proves a very good fit for the real values.

#### 3.1.6. Pelvic ST

The coefficients of the model selected by stepwise regression for pelvic ST (P_ST) as a dependent variable are presented in Table 10. The value of statistic F = 21.41 and the corresponding level of the *p*-value (*p* < 0.001) confirm a statistically significant linear relationship. 

Also, for the hind limbs, as for the forelimbs, the obtained adjusted coefficient of determination R^2^ = 0.556 proves a moderate fitting of the model to the real data.

According to the coefficients in Table 10, the regression equation takes the following form (Model 6):$$P_{ST}=-0.089+0.009\cdot BW+0.008\cdot WH-9.088^{-5}\cdot BW\cdot WH$$

The *BW* and *WH* have a significant effect on pelvic *ST*, even with an increased minimal average. To compare actual data with the model’s predictions, we can visualise the correlation scatterplot of P_ST for the observed values predicted by the model. As shown in Figure 6, the obtained model proves a poor fit for the real values.

## 4. Discussion

The effect of morphometric measures on GRFs, their combined effect, and possible interactions in healthy dolichomorph and mesomorph dogs were assessed in this prospective study. Moreover, the differences in these relationships in the two morphological groups were considered to achieve objective reference values in the considered canine morphotype. 

The force plate analysis is an objective, quantifiable, and repeatable technique used to assess normal and abnormal gait in dogs [8,13]. The GRFs analysis allows for detecting gait symmetry and limb loading. Peak vertical force and VI, when recorded through the use of a pressure plate, were found to be reliable indicators of clinical lameness in dogs and have therefore been used as a method to assess clinical outcomes following surgical treatment [14,15,16,17], conservative treatment for OA [18], or evaluate the correspondence between the radiographic and/or clinical findings and the load borne by the examined limb [19,20]. 

The PVF and VI are unquestionable measures of limb function. However, a substantial variability related to the specific breeds’ morphometric features due to the differences in body conformation and size between the breeds and between different subjects of the same breed was described in the literature [1,6]. These make poorly reliable comparisons between groups despite the similar breed or body conformation and weight. In the present study, two different canine morphological types were considered. Several breeds belonging to the respective morphological types were included, hunting and guard breeds for mesomorphs and many greyhound breeds for dolichomorphs.

In the present study, all dogs were evaluated at walking speed, because acquiring valid trials at trot requires more repetitions, reducing the compliance of examined subjects [21]. 

In nature, phenomena can only be explained by the influence of several elements simultaneously and can be studied by evaluating the interactions between the variables taken into consideration. In the performed analyses, we preferred not to use the normalised GRF measures as proposed in the literature [2,6], since our goal was not to compare PVF, VI, and ST values among different subjects but to assess the relationships between the GRF measurements and the morphometric measures, hence the decision to set up the statistical analysis by comparing a single response variable with several predictors using the multiple linear regression. 

Our results highlight how the *BW* affects both groups’ pelvic PVF (2.518 N for each Kg of *BW*). Otherwise, in dolichomorphs, the *BW* affect more the *T_PVF* (2.19 N for each kg of *BW*), than *P_PVF*. Meanwhile, in mesomorphs, *T_PVF* increases further by 1.14 N for each kg of *BW* compared to the dolichomorphs group.

Moreover, in mesomorphs, the interaction between *BW* and *WH* has an additional mean increase effect on both *T_PVF* and T_VI, with a mean increase directly proportional to the increase of *BW* and/or *WH*. This phenomenon is linked, in our opinion, to the influence of the more developed thoracic muscular component, the neck, and the weight of the head in the mesomorphs. Also, the *V* showed a positive correlation with an increasing average value on thoracic and pelvic PVF, respectively. 

These data, in our opinion, are related to the gait of the dogs in the mesomorph group, all belonging to guard breeds with body shapes more long than high. Similarly, the VI reflects the effect of the interaction of *BW* and *WH* in the forelimbs but only of *BW* in the hindlimbs. 

This finding partly repeats what is already known and reported in the literature, but our analysis showed that both *BW* and the relation to *V* and/or *WH* influence the GRFs [1,2,3,22].

For mesomorphs, the increase in *WH* was inversely proportional to VI in hind and forelimbs, while it is positively correlated to the increase in *BW* only in forelimbs. This could be related, once more, to the more cranially located centre of gravity of these subjects yielding a greater weight distribution on the forelimbs.

In our study, the influence of the morphometric measures on the ST was minimal, even if it was statistically significant. The ST parameter, therefore, frequently shows unpredictable results from the variables considered in our study.

Moreover, it is interesting to note how the lengths of the femur and humerus were not the best explanatory variables for the GRFs and were progressively deleted by the regression models.

## 5. Conclusions

In conclusion, our results show how the GFRs are influenced by morphometric measures not so much as a single contribution but through the interaction between them. The interaction between *BW*, *WH*, and *V* significantly influenced GFRs, with a greater unit increase for mesomorphs.

Statistical models reported in the available literature only partially explain the influence of morphometric measures on GFRs. The approach to the statistical processing of data obtained from gait analysis cannot disregard the evaluation of interactions between variables.

Despite various proposed normalisation formulas, comparing GFRs in dogs belonging to different breeds and even within the same breed is not possible, as subjective variabilities make this comparison difficult.

According to our results, the comparison must not be made referring to the breeds but to the canine morphological type.

## Figures and Tables

**Figure 1 vetsci-09-00325-f001:**
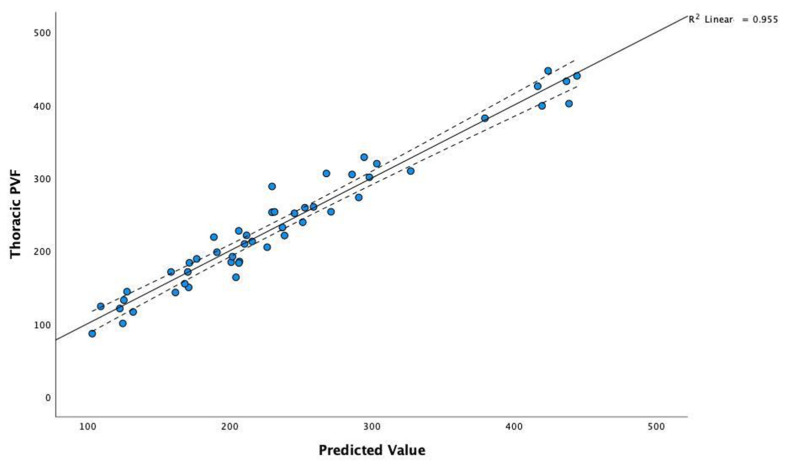
Correlation scatterplot of thoracic PVF for the observed values and those predicted by the model.

**Figure 2 vetsci-09-00325-f002:**
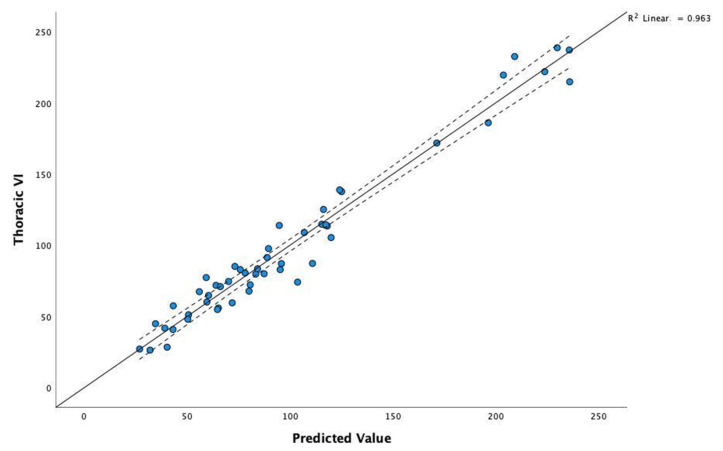
Correlation scatterplot of Thoracic VI for the observed values and those predicted by the model.

**Figure 3 vetsci-09-00325-f003:**
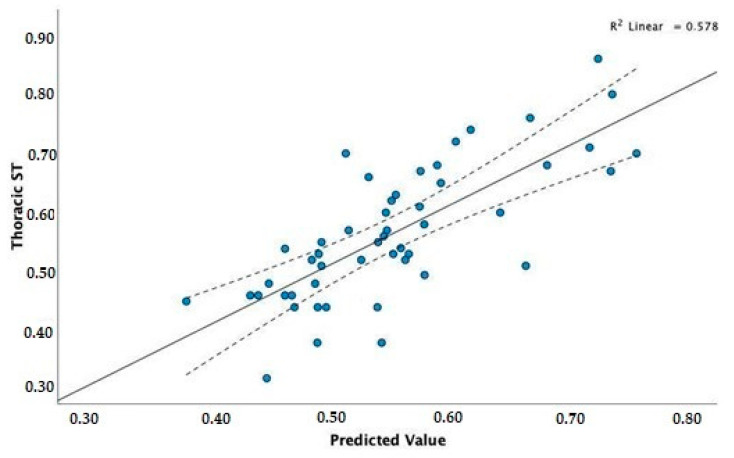
Correlation scatterplot of thoracic ST for the observed values and those predicted by the model.

**Figure 4 vetsci-09-00325-f004:**
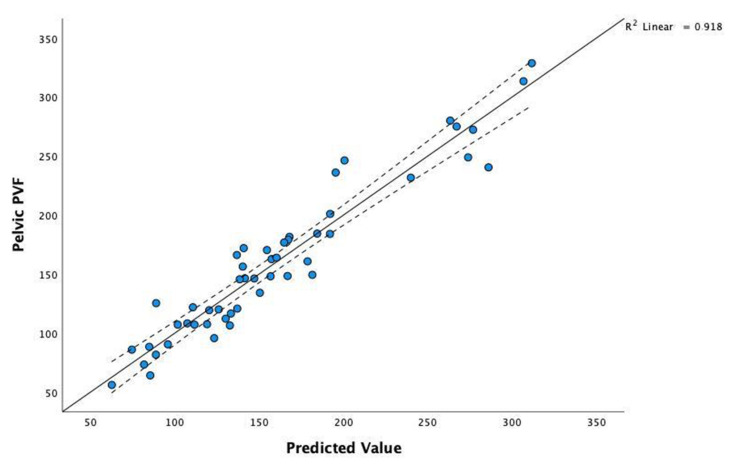
Correlation scatterplot of pelvic PVF for the observed values and those predicted by the model.

**Figure 5 vetsci-09-00325-f005:**
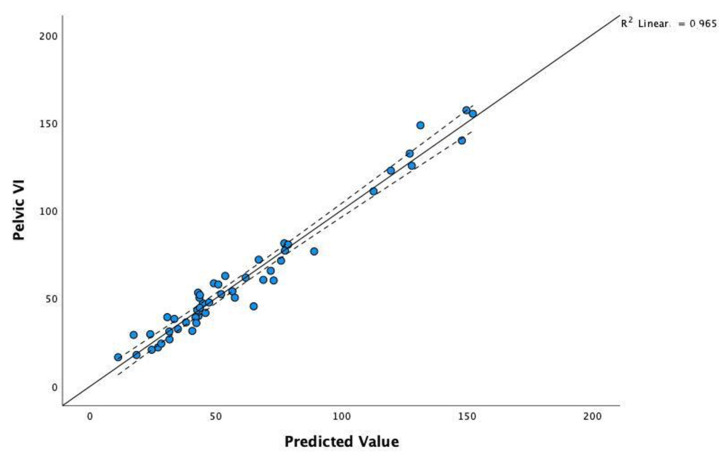
Correlation scatterplot of pelvic VI for the observed values and those predicted by the model.

**Figure 6 vetsci-09-00325-f006:**
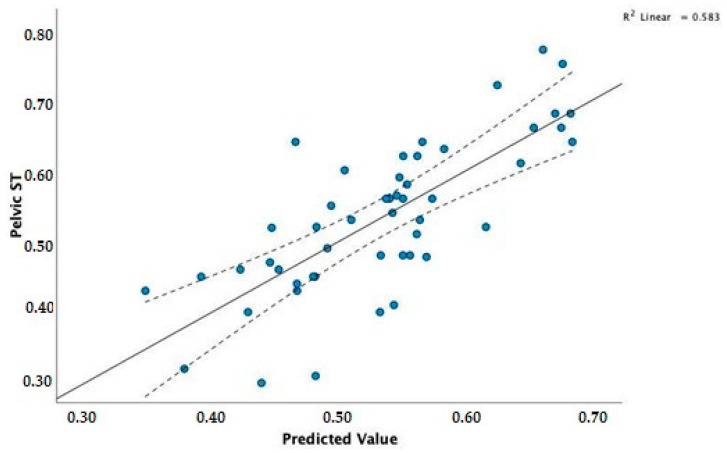
Correlation scatterplot of pelvic ST for the observed values and those predicted by the model.

**Table 1 vetsci-09-00325-t001:** Morphological type distribution according to sex.

Morphological Types	Male	Female
Dolichomorph	9	5
Mesomorph	16	20

**Table 2 vetsci-09-00325-t002:** Distribution of canine breed in the dolichomorphs group.

Breed	Number
Azawakh	1
Borzoi	3
Greyhound	1
Irish Wolfhound	4
Saluki	2
Whippet	3
TOTAL	14

**Table 3 vetsci-09-00325-t003:** Distribution of canine breed in the mesomorphs group.

Breed	*n*	Breed	*n*
Akita Inu	1	Dogue de Bordeaux	1
American Akita	1	Golden Retriever	1
American Staffordshire	1	Great Dane	1
Bobtail	1	Hovawart	1
Border collie	1	Labrador Retriever	3
Boxer	1	Pyrenean Mastiff	2
Bullmastiff	1	Saarloos	1
Central Asian Shepherd dog	1	Siberian Husky	1
Corso	1	Tibetan Mastiff	2
Czechoslovakian wolfdog	3	Tibetan mastiff Shepherd	1
Drahthaar	1	Weimaraner	3
Dogo Argentino	1	TOTAL	36

**Table 4 vetsci-09-00325-t004:** Morphometric measurement in cm expressed as median (range).

	Withers Height (*WH*)	Length of the Trunk (*L*)	Humerus (O)	The Femur (*F*)
Dolichomorphs (range)	73.25 cm (50–91.5)	55.75 cm (36–69)	19 cm (13.5–28)	21.5 cm (15–30)
Mesomorphs (range)	63.25 cm (46–88.5)	47.25cm (36–71.5)	18 cm (11–24.5)	22 cm (15.5–31.5)

**Table 5 vetsci-09-00325-t005:** Coefficients of stepwise backward regression for the dependent variable Thoracic PVF.

Model	Coefficients	Standard Error	T	*p*-Value
Intercept	55.32	13.63	4.058	0.000
Dummy Mesomorphic = 1	−344.25	135.91	−2.533	0.015
Body Weight	2.19	0.99	2.210	0.032
Int_D_BW	1.14	0.43	2.659	0.011
Int_D_V	289.62	115.92	2.498	0.016
Int_BW_WH	0.03	0.01	3.157	0.003

Adjust. R^2^ = 0.949 – F-statistic = 184.92 (*p* < 0.001).

**Table 6 vetsci-09-00325-t006:** Coefficients of stepwise backward regression for the dependent variable thoracic VI.

Model	Coefficients	Standard Error	T	*p*-Value
Intercept	6.77	4.73	1.432	0.159
Int_D_BW	0.65	0.21	3.138	0.003
Int_D_WH	−0.72	0.21	−3.420	0.001
Int_D_L	0.54	0.31	1.747	0.087
Int_BW_WH	0.03	0.001	26.278	0.000

Adjust. R^2^ = 0.959 – F-statistic = 291.11 (*p* < 0.000).

**Table 7 vetsci-09-00325-t007:** Coefficients of stepwise backward regression for the dependent variable thoracic ST.

Model	Coefficients	Standard Error	T	*p*-Value
Intercept	0.083	0.074	1.119	0.269
Withers Height	0.007	0.001	6.962	0.000
Int_D_BW	0.003	0.001	2.580	0.013
Int_D_WH	−0.002	0.001	−2.067	0.044

Adjust. R^2^ = 0.550 – F-statistic = 20.96 (*p* < 0.001).

**Table 8 vetsci-09-00325-t008:** Coefficients of stepwise backward regression for the dependent variable pelvic PVF.

Model	Coefficients	Standard Error	T	*p*-Value
Intercept	27.658	8.407	3.290	0.002
Dummy Mesomorphic = 1	−385.559	118.710	−3.248	0.002
Body Weight	2.518	0.669	3.766	0.000
Int_D_V	336.132	1,040,092	3.229	0.002
Int_BW_WH	0.014	0.007	1.883	0.066

Adjust. R2 = 0.911 – F-statistic = 126.7 (*p* < 0.001).

**Table 9 vetsci-09-00325-t009:** Coefficients of stepwise backward regression for the dependent variable pelvic VI.

Model	Coefficients	Standard Error	T	*p*-Value
Intercept	−13.488	2.845	−4.741	0.000
Body Weight	2.198	0.061	35.984	0.000
Int_D_WH	−0.183	0.035	−5.264	0.000

Adjust. R^2^ = 0.964 – F-statistic = 648.53 (*p* < 0.001).

**Table 10 vetsci-09-00325-t010:** Coefficients of stepwise backward regression for the dependent variable pelvic ST.

**Model**	**Coefficients**	**Standard Error**	**T**	***p*-Value**
Intercept	−0.089	0.157	0.563	0.576
Body Weight	0.009	0.003	2.768	0.008
Withers Height	0.008	0.003	3.118	0.003
Int_BW_WH	−9.088 × 10^−5^	0.000	−1.941	0.058

Adjust. R^2^ = 0.556 – F-statistic = 21.41 (*p* < 0.001).

## Data Availability

Not applicable.

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
