# Peer review of "Relationship between Ground Reaction Forces and Morpho- Metric Measures in Two Different Canine Phenotypes Using Regression Analysis"

_vetsci, 2022, doi:10.3390/vetsci9070325_

Round 1
Reviewer 1 Report
the study is well designed and of great interest to any orthopaedic surgeon and/or specialist in sports medicine; the concept of elaborating base-line physiologic gait data in dogs of different conformation is sound and quite novel. However, the inclusion criteria for the study subjects remains to be clarified; that is, the allocation of dogs of different breeds in cohorts, resulting in a division into two distinct groups must be objective, reproducible and inambigous; the methods and the mathematical elaboration of data are statistically well documented; however, this reviewer feels not sufficiently competent to evaluate the statistics. The resulting conclusions should be stated clearly in terms of evidence, that is, numbers and data "in the Author's opinion..." seems too weak as statement unless the results are ambiguous. I am attaching a Word file of the manuscript with recommendations and comments.

Author Response
As corresponding, the authors (AU) thank the Reviewer for the important and valuable comments that helped to improve the quality of the manuscript. The AU have carefully considered every reviewer’s recommendation and the manuscript has been revised according to their suggestions. Please see the attachment.
Your Sincerely
Dr. Federica Aragosa

Reviewer 2 Report
Thank you for this manuscript, which represents without any doubt a considerable amount of work. This study aims at evaluating kinetics in healthy dogs, with a special focus on morphometric features. Very few original information is provided and clinical impact, at this stage, might be very limited. In my opinion, as results are still interesting, a substantial work is imperative to highlight the originality and significance of the content.
It is sometimes a bit confusing when going through the paper, particularly in the results section and, it is thus easy to loose the plot. I also strongly recommend an extensive editing of English language and style.
Below you can find my suggestions and corrections.
Abstract:
Line 21: Please reformulate. Variability of what?
Line 26: GFR has not been defined yet
Line 28: "to assess the impact...", it is a repetition of the sentence Line 22. It can be deleted.
Line 32: Sentence should be ended after "same breed". New sentence starts with "However...".
Introduction
Line 41: with musculoskeletal disease
Lin 46: Please reformulate or complete the sentence
Line 47: 7 different breeds
Line 48: concluded
Line 49: authors suggested
Line 53: in the data?
Lines 55-56: Please reformulate and try to use the same tense throughout the section
Line 60: and colleagues (2010)
Line 65: literature background, we hypothesize...
Line 66: do not act as a single variable
Line 71: either delete "in healthy dolicomorphs and mesomorphs dogs" as it is stated for a), or report it in the same way as a)
Materials and Methods
Line 75: members?
Line 80: related to instead of belonging. Please reformulate this sentence to be more precise and clear
Line 85: assigned to 1 of the 2 groups
Line 93: highest
Line 94: scapulae
Lines 101-102: Please gather all this information in one sentence
Line 104: let to walk free
Results
I am not that qualified to evaluate in depth the statistics. In my opinion, it is very unpleasant to go through this section which is very repetitive, with lot of abbreviations. It is sometimes better when dealing with such a huge amount of data to gather all major information in 1 or 2 tables, to which you refer in your manuscript. It avoids to have too long results sections.
Line 169: Error in the Table 1 regarding males (n=25)
Line 172: Greyhound
Line 173: this sentence lacks a verb
Line 175: Central Asian Shepherd dog
Line 185: p<0.001
Line 194: Is "morphological typology" the right term? It seems that it doesn't refer to what is meant
Line 196: "the canine group" should be reformulated to be more specific. In that way, it could be interpreted as a "species group".
Line 240: Incomplete sentence?
Line 262: morphological typology, please refer to comment on Line 194
Line 264: Canine group, please refer to comment on Line 196
Line 285: shows
Discussion
Line 315: Sentence summarizing major data obtained in the study should be added
Lines 325: Please reformulate this sentence.
Line 327: poorly
Lines 330: this -> each?
Line 332: Please reformulate this sentence.
Line 343: delete "as well"
Line 345: Please reformulate this sentence.
Line 353: fells?
Line 363: delete "were"
Lines 365-367: Please reformulate these sentences. This could be shortened and more straightforward.
Author Response

(The authors gave the same response as above.)

Round 2
Reviewer 2 Report
Thank you for the consideration you gave to my suggestions. The manuscript is well improved but, to my humble opinion, still needs rigorous English editing.
Below, you can find some suggestions. However, these can't preclude you from taking advise from a native English speaker or from a professional editing service.
- "Force plate" should always be written in the same way (force-plate vs force plate)
- "Ground reaction forces" should always be written in the same way (Ground Reaction Forces, Ground reaction forces, ...)
Line 77: Peak and not Peck
Line 78: "showed that " instead of "conclude"
Line 85: investigated and proposed or investigated and suggested
Line 97: hypothesized
Line 107: taking into account
Line 188: 30 minutes
Line 199: s2
Line 220: estimated
Line 220: 6 different models obtained by 6 stepwise
Line 221: "Canine group", please refer to my suggestion in Review 1
Line 231: "Canine group", please refer to my suggestion in Review 1
Line 231: was used
Line 404: average decrease
Line 447: between different subjects of the same breed
Line 454: delete the dot before (21)
Line 500: relationships
Line 505-508: Please reformulate this paragraph. This is not understandable and full of grammatical errors. Please reformulate properly.
Line 509-511: positive effect. These sentences are also not clearly understandable and needs English editing. Please reformulate properly.
Line 543: as subjective
Line 545: but to the canine
Author Response
As corresponding, the author (AU) thank the Reviewer for the important and valuable comments that helped to improve the quality of the manuscript. The AU have carefully considered every reviewer’s recommendation and the manuscript has been revised according to their suggestions.
Unfortunately, we are unable to identify the 2 sentences that you signaled to us in the text, as the line numbers indicated did not correspond to the manuscript latest version. However, we have proceeded to an extensive editing of English language. We hope that now the paper has all the necessary qualifications for publication in Veterinary Sciences.
Your Sincerely
Dr. Federica Aragosa